# DA7R: A 7-Letter Zip Code to Target PDAC

**DOI:** 10.3390/pharmaceutics15051508

**Published:** 2023-05-16

**Authors:** Sofia Parrasia, Andrea Rossa, Nicola Roncaglia, Andrea Mattarei, Claudia Honisch, Ildikò Szabò, Paolo Ruzza, Lucia Biasutto

**Affiliations:** 1Department of Biology, University of Padova, Viale G. Colombo 3, 35131 Padova, Italy; sofia.parrasia@unipd.it (S.P.); ildiko.szabo@unipd.it (I.S.); 2Department of Chemical Sciences, University of Padova, Via F. Marzolo 1, 35131 Padova, Italy; andrea.rossa@unipd.it (A.R.);; 3CNR Institute of Biomolecular Chemistry, Padua Unit, Via F. Marzolo 1, 35131 Padova, Italy; c.honisch@icb.cnr.it; 4Department of Pharmaceutical and Pharmacological Sciences, University of Padova, Via F. Marzolo 5, 35131 Padova, Italy; andrea.mattarei@unipd.it; 5CNR Neuroscience Institute, Padua Unit, Viale G. Colombo 3, 35131 Padova, Italy

**Keywords:** pancreatic ductal adenocarcinoma (PDAC), A7R peptide, neuropilin-1 (NRP-1), vascular endothelial growth factor receptor-2 (VEGFR2), PAPTP

## Abstract

Pancreatic ductal adenocarcinoma (PDAC) is the most common type of pancreatic cancer, and is among the most aggressive and still incurable cancers. Innovative and successful therapeutic strategies are extremely needed. Peptides represent a versatile and promising tool to achieve tumor targeting, thanks to their ability to recognize specific target proteins (over)expressed on the surface of cancer cells. A7R is one such peptide, binding neuropilin-1 (NRP-1) and VEGFR2. Since PDAC expresses these receptors, the aim of this study was to test if A7R-drug conjugates could represent a PDAC-targeting strategy. PAPTP, a promising mitochondria-targeted anticancer compound, was selected as the cargo for this proof-of-concept study. Derivatives were designed as prodrugs, using a bioreversible linker to connect PAPTP to the peptide. Both the retro-inverso (DA7R) and the head-to-tail cyclic (cA7R) protease-resistant analogs of A7R were tested, and a tetraethylene glycol chain was introduced to improve solubility. Uptake of a fluorescent DA7R conjugate, as well as of the PAPTP-DA7R derivative into PDAC cell lines was found to be related to the expression levels of NRP-1 and VEGFR2. Conjugation of DA7R to therapeutically active compounds or nanovehicles might allow PDAC-targeted drug delivery, improving the efficacy of the therapy and reducing off-target effects.

## 1. Introduction

Pancreatic cancer is one of the most common and fearsome malignant tumors of the digestive system. Worldwide, nearly half a million people were diagnosed with pancreatic cancer in 2020, accounting for about 3% of all cancers (https://www.cancer.net/cancer-types/pancreatic-cancer/statistics, accessed on 3 April 2023). Despite intensive research, it is still among the malignancies with the most unfavorable prognosis [1,2]. The general 5-year survival rate in the US is about 11%. The corresponding figure for nonmetastatic invasive breast cancer, for example, is 90% (https://www.cancer.net/cancer-types/breast-cancer/statistics, accessed on 3 April 2023). Pancreatic ductal adenocarcinoma (PDAC) [3,4] is the most common and lethal type of pancreatic cancer, accounting for about 90% of total cases. 

The challenges posed by PDAC derive from multiple factors, including lack of symptoms and consequent late diagnosis, early metastasis, difficulties in surgical resection and, notably, resistance to pharmacological treatments [5]. The tumors generally present concurrent mutations in multiple, poorly druggable driver genes, the “signature” ones being *KRAS*, *P53*, *CDKN2A,* and *SMAD4* [6,7]. The presence of a complex, immunosuppressive stroma, which may account for up to 80% of the tumor mass, plays a major role in cancer progression and constitutes a unique and tight barrier impeding drug delivery to tumor cells [8,9,10].

Many approaches are under investigation with the aim of improving PDAC therapy (rev: [3,11]). Using peptides is one of the possible delivery strategies to achieve tumor targeting. Peptides are extremely versatile thanks to the numerous possible combinations of natural (and also unnatural) amino acids in the sequence, and can be selected by phage display methods for their ability to recognize a desired target, such as surface proteins (over)expressed by cancer cells [12,13,14]. One may thus advantageously interfere with cellular signaling. Furthermore, conjugation of a suitable peptide to therapeutically active compounds or to nanovehicles may allow tumor-specific delivery of the “cargo”, improving the efficacy of the therapy and reducing off-target side effects. Receptors present in the tumor vasculature and/or overexpressed by tumor cells thus constitute an interesting target to deliver drugs/nanovehicles specifically to the tumor.

Surveying the literature, we noticed the promising studies with peptide ATWLPPR, named “A7R” [15,16]. Since the peptide undergoes rather rapid proteolytic degradation [17,18], more stable retro-inverso (DA7R) and cyclic (cA7R) variants have been produced and successfully tested [19,20]. The use of D-amino acids, or cyclization, confers resistance to proteolysis, extending the lifetime of the peptide, an important feature in drug delivery [21,22]. 

A7R peptide binds to both neuropilin-1 (NRP-1) and vascular endothelial growth factor receptor-2 (VEGFR2/KDR), disrupting their receptor/coreceptor interaction, and impacting, therefore, on angiogenesis and tumor growth [23,24,25]. Molecular docking studies of A7R with the crystal structure of VEGFR2 suggested that the peptide deeply inserts between the two domains of VEGFR2; interactions mainly involve the Trp residue, but also Leu and Pro, while Arg locates at the outer edge of the binding pocket. Binding of A7R to NRP-1, on the other hand, mainly involves interactions of Arg with the b1 domain of NRP-1 [16,19]. The C-terminal sequence LPPR, and Arg particularly, are crucial for the inhibitory effect of A7R on VEGF165 binding to NRP-1 [26]. Protease-resistant A7R variants (DA7R, cA7R) were demonstrated to retain affinity for both VEGFR2 and NRP-1, as measured by surface plasmon resonance, and simulated by molecular docking studies [19,20]. These results seem to indicate that the presence of a free Arg side chain (despite the use of a retro-inverso sequence or the cyclization involving the C-terminal) is sufficient to preserve peptide affinity to NRP-1. NRP-1 engagement prompts the activation of a macropinocytosis-like, GIPC1/synectin-dependent process, with the consequent internalization of the receptor complex and extracellular fluid [27,28]. A7R variants and constructs comprising them have been used for therapeutic, imaging, and drug/nanovehicle delivery purposes (revs: [16,29]), mostly with models of glioma, possibly because neuropilin was originally identified as a regulator of neurogenesis. To the best of our knowledge, they have never been investigated as a strategy to target PDAC. On the other hand, NRP-1 and VEGFR2 are variably expressed by PDAC vessels and cells [30,31,32,33,34,35,36]. Indeed, neuropilins have been advocated as “ideal therapeutic targets against PDAC” [37]. 

On these bases, we undertook this study to prove in principle that A7R can be used to deliver drugs to PDAC cells. 

Among the various cargoes that could be tested for delivery to the cells, we opted for PAPTP, one of the promising mitochondria-targeting compounds our group is currently researching. Mitochondria are emerging as a crucial target for cancer therapy [38,39,40,41]. A family of mitochondria-targeted chemotherapeutics has been developed starting from the psoralenic compound PAP-1, a well-known and specific inhibitor of the Kv1.3 voltage-gated potassium channel [42]. Thanks to the conjugation with a lipophilic cation (triphenyphosphonium), which addresses the molecule to the mitochondrial matrix [43], the resulting derivatives (PAPTP and PCARBTP) bind to and inhibit the Kv1.3 channel residing in the inner mitochondrial membrane, causing oxidative stress and selectively killing cancer cells while sparing normal ones [44]. Mitochondrial targeting is essential in determining the anticancer effectiveness of the compounds, since PAP-1 is much less effective. 

What makes PAPTP particularly interesting is its effectiveness in killing a variety of cancer cells, including those deriving from glioma [45] and PDAC [46], i.e., tumors still lacking effective therapeutic options. Since these drugs act at the level of mitochondria, their action is independent of the status of upstream oncoproteins, such as Bcl-2 or p53 [44].

In this work, we synthesized different variants of the PAPTP-A7R conjugate. PAPTP was modified with the addition of a short linker to one of the phenyl rings of the triphenylphosphonium group (PAPTPL), to allow conjugation with the peptide in a position not interfering with binding to and inhibition of Kv1.3, and at the same time to ensure the suitable stability of the resulting derivative [47]. PAPTPL was connected to DA7R or cA7R via a bioreversible linker, allowing release of the active principle according to the “prodrug” concept (Figure 1). The different conjugates were preliminarily characterized for their stability in blood and for their hemolytic activity. Cell uptake of a fluorescent FITC-DA7R conjugate, as well as of a PAPTP-DA7R conjugate (PAPTPL-I-DA7R), was finally evaluated in PDAC cell lines. 

## 2. Materials and Methods

### 2.1. Chemistry

#### 2.1.1. Materials and Instruments

Fmoc-protected amino acids, preloaded 2-Cl-Trytil resin, and coupling reagents (HBTU and HOBt) were purchased from Iris Biotech (Marktredwitz, Germany). Other reagents and solvents were purchased from Sigma Aldrich (Milan, Italy), and were used as received. 

Flash chromatography was performed on silica gel (Macherey-Nagel 60, 230–400 mesh granulometry (0.063–0.040 mm), Fisher Scientific Italia, Segrate (MI), Italy) under air pressure.

Preparative HPLC was performed using a Shimadzu LC-8 system (Shimazdu, Kyoto, Japan) with a C18 column (Vydac 218TP1022, 10 µm, 250 × 22 mm). The column was perfused at a flow rate of 12 mL/min with a mobile phase consisting of eluent A (0.05% TFA in water) and B (0.05% TFA in 9:1 *v*/*v* acetonitrile:water); eluent B was increased over time following a linear gradient.

Analytical HPLC analyses were performed on a Shimadzu LC-10 instrument fitted with a Jupiter C18 column (10µm, 250 × 4.6 mm Phenomenex, Torrance, CA, USA) using the above eluent system (eluents A and B), flow rate of 1 mL/min, detection at 216 nm. 

^1^H and ^13^C nuclear magnetic resonance (NMR) spectra were recorded with a Bruker 500 Avance III operating at 500 MHz (for ^1^H NMR) and 126 MHz (for ^13^C NMR). Chemical shifts (δ) are given in parts per million (ppm) relative to the signal of the solvent. The following abbreviations are used to indicate multiplicities: d, doublet; t, triplet; m, multiplet.

Electrospray ionization-mass spectrometry (ESI-MS) analysis was carried out with a Mariner mass spectrometer (PerSeptive Biosystem, Framingham, MA, USA) or a 1100 Series Agilent Technologies system (Agilent Technologies, Milan, Italy). The ESI source operated in full-scan positive ion mode, applying the following ESI parameters: nebulizer pressure 20 psi, dry gas flow 5 L/min, dry gas temperature 325 °C. The flow rate was 0.05 mL/min.

Microwave-assisted syntheses were performed in a CEM Discover^®^ monomode reactor (CEM Srl, Cologno Al Serio (BG), Italy) with the temperature monitored by a built-in infrared sensor.

#### 2.1.2. Peptide Synthesis

Peptides were synthesized by manual solid phase using Fmoc chemistry in 0.06 mmole scale using a preloaded (L-Arg or D-Ala) 2-Cl-Trt-resin. HBTU/HOBt activation employed a three-fold molar excess (0.24 mmol) of Fmoc-amino acids in DMF (N,N-Dimethylformamide) for each coupling cycle, unless otherwise stated. Coupling time was 40 min. Fmoc deprotection was performed with 20% piperidine. Coupling yields were monitored on aliquots of peptide-resin by evaluation of Fmoc displacement. The cleavage of peptide from the resin was performed by treatment with 1% TFA in DCM (Dichloromethane) (4 × 5 mL) followed by neutralization with pyridine, to preserve the side chain protecting groups. Deprotection of peptides was achieved by treatment with a TFA-triisopropylsilane-H_2_O (96:3:1 *v*/*v*) mixture for 45 min at room temperature. Pra-A7R peptide was cyclized in dilute DMF solution (1 mM) by the addition of 2 eq. of PyBOP in the presence of HOBt (2 eq.) and DIPEA (4 eq.). Peptides were purified by preparative reversed-phase HPLC. The fractions containing the desired products were collected and lyophilized to constant weight. All peptides showed less than 1% impurities. Molecular weights of compounds were determined by ESI-MS on a Mariner mass spectrometer. The mass was assigned using a mixture of neurotensin, angiotensin, and bradykinin at a concentration of 1 pmol/μL as external standard.

#### 2.1.3. Synthesis of FITC-DA7R 

FITC-DA7R was synthesized starting from the H-DA7R-2Cl-Trt resin by reaction with fluorescein-5-isothiocyanate (FITC) (2 eq.) in the presence of DIPEA (4 eq.) in DMF for 45 min. Removal of protecting groups and cleavage of the peptide from the resin by treatment with TFA-triisopropylsilane-H_2_O mixture yielded the final desired peptide, which was purified and characterized as reported above (see also Appendix A). 

#### 2.1.4. Synthesis of PAPTPL-I-DA7R

PAPTPL-I was synthesized as described in [47]. PAPTPL-I (30.0 mg, 27.9 µmol, 1 eq.), HATU (10.6 mg, 27.9 µmol, 1 eq.), and HOAt (3.8 mg, 28 µmol, 1 eq.) were dissolved in DMF (1.0 mL). After the addition of DIPEA (9.7 µL, 56 µmol, 2 eq.), the mixture was stirred for 1 h at room temperature and then transferred onto 65 mg of H_2_N-DA7R-2-Cl-Trytil resin (1 eq.). DMF (1.0 mL) was added, and the suspension was stirred at room temperature overnight. The resin was successively washed with DMF, DCM, and dried under reduced pressure. PAPTPL-I-DA7R was side chain deprotected and detached from the resin by treatment with TFA-H_2_O-TIPS 95:2:3 *v*/*v*, for 3 h. At the end of the treatment, most of the TFA was stripped off with N_2_ and the peptide derivative was precipitated by addition of Et_2_O. The solid was collected by centrifugation and removal of the supernatant. The crude product was purified by preparative HPLC (linear gradient of eluent B from 55 to 80% in 35 min) to yield PAPTPL-I-DA7R (7.2 mg, 3.7 µmol, 13% yield) as a white powder. ESI-MS (*m*/*z*): 891.9 (calculated 891.4) [M + H]^2+^.

#### 2.1.5. Synthesis of PAPTPL-Teg-DA7R and PAPTPL-Teg-cA7R

##### PAPTPL-Teg-N_3_ Synthesis


*Synthesis of 4-nitrophenyl (14-azido-3,6,9,12-tetraoxatetradecyl)carbamate (*
**1**
*)*


The compounds 14-azido-3,6,9,12-tetraoxatetradecan-1-amine (500 mg, 1.91 mmol, 1 eq.), 4-dimethylaminopyridine (DMAP) (466 mg, 3.81 mmol, 2 eq.), and bis(4-nitrophenyl)carbonate (640 mg, 2.10 mmol, 1.1 eq.) were dissolved in CH_3_CN (7.5 mL). The mixture was stirred for 3 h at 50 °C, then the solvent was removed under reduced pressure. The concentrated mixture was dissolved in DCM (50 mL) and washed with 0.5 M HCl (50 mL). The aqueous phase was extracted with DCM (3 × 50 mL) and the total organic phase was dried over sodium sulfate. The solvent was removed under reduced pressure and the crude product was purified by flash chromatography (DCM:acetone, 9:1 as eluent) to afford **1** (484 mg, 1.13 mmol, 59% yield) as a transparent oil (Figure 1). ^1^H NMR (500 MHz, CDCl_3_) δ 8.25–8.18 (m, 2H), 7.33–7.27 (m, 2H), 6.12–6.00 (m, 1H), 3.69–3.60 (m, 16H), 3.48–3.43 (m, 2H), 3.38–3.33 (m, 2H). ^13^C NMR (126 MHz, CDCl_3_) δ 156.17, 153.39, 144.76, 125.16, 122.09, 70.76, 70.71, 70.64, 70.62, 70.38, 70.09, 69.71, 50.71, 41.25. ESI-MS (*m*/*z*): 428.1 (calculated 428.2) [M + H]^+^, 445.2 (calculated 445.2) [M + NH_4_]^+^, 450.1 (calculated 450.2) [M + Na]^+^.


*Synthesis of (4-(3-(4-(((14-azido-3,6,9,12-tetraoxatetradecyl)carbamoyl)oxy)phenyl)propoxy)phenyl)(3-(4-(4-((7-oxo-7H-furo [3,2-g]chromen-4-yl)oxy)butoxy)phenyl)propyl)diphenylphosphonium 2,2,2-trifluoroacetate (PAPTPL-Teg-N_3_)*


PAPTPL was synthesized as described in [47]. PAPTPL (321 mg, 350 µmol, 1 eq.), intermediate **1** (299 mg, 700 µmol, 2 eq.) and DMAP (85.5 mg, 700 µmol, 2 eq.) were dissolved in CH_3_CN (3.0 mL). The mixture was stirred for 95 h at 50 °C, then the solvent was removed under reduced pressure. The concentrated mixture was dissolved in DCM (50 mL) and washed with 0.5 M HCl (50 mL). The aqueous phase was extracted with DCM (2 × 50 mL) and the total organic phase was dried over sodium sulfate. The solvent was removed under reduced pressure and the crude product was purified by flash chromatography (eluents: DCM:acetone, 9:1 *v*/*v* to remove impurities, then DCM:MeOH, 8:2 *v*/*v* to get the desired product) to yield PAPTPL-Teg-N_3_ (392 mg, 326 µmol, 93% yield) as a slightly yellow viscous oil (Figure 1). ^1^H NMR (500 MHz, CDCl_3_) δ 8.12 (d, J = 9.8 Hz, 1H), 7.76 (t, J = 7.2 Hz, 2H), 7.67–7.54 (m, 9H), 7.53–7.46 (m, 2H), 7.15 (d, J = 8.3 Hz, 2H), 7.12–7.06 (m, 3H), 7.06–6.99 (m, 4H), 6.95 (d, J = 1.6 Hz, 1H), 6.79 (d, J = 8.3 Hz, 2H), 6.18 (d, J = 9.8 Hz, 1H), 5.88 (t, J = 5.4 Hz, 1H), 4.52 (t, J = 5.9 Hz, 2H), 4.07–3.98 (m, 4H), 3.70–3.57 (m, 16H), 3.47–3.39 (m, 2H), 3.35 (t, J = 4.9 Hz, 2H), 3.27–3.16 (m, 2H), 2.82–2.72 (m, 4H), 2.15–1.95 (m, 6H), 1.94–1.82 (m, 2H). ^13^C NMR (126 MHz, CDCl_3_) δ 164.49 (d, J_C-P_ = 3.0 Hz), 161.34, 158.36, 157.63, 155.05, 152.73, 149.53, 149.04, 145.00, 139.50, 137.88, 135.53 (d, J_C-P_ = 11.5 Hz), 135.10 (d, J_C-P_ = 2.8 Hz), 133.32 (d, J_C-P_ = 9.9 Hz), 131.86, 130.51 (d, J_C-P_ = 12.5 Hz), 129.84, 129.29, 121.74, 118.80 (d, J_C-P_ = 86.6 Hz), 116.84 (d, J_C-P_ = 13.7 Hz), 114.72, 113.30, 112.47, 106.99 (d, J_C-P_ = 93.7 Hz), 106.72, 105.27, 93.82, 72.63, 70.75, 70.71, 70.67, 70.64, 70.62, 70.37, 70.06, 69.93, 67.68, 67.42, 50.73, 41.12, 34.99 (d, J_C-P_ = 16.6 Hz), 31.37, 30.42, 27.00, 26.00, 24.56 (d, J_C-P_ = 3.7 Hz), 21.67 (d, J_C-P_ = 52.7 Hz). ESI-MS (*m*/z): 1091.6 (calculated 1091.5) [M]^+^, 557.2 (calculated 557.2) [M + Na]^2+^, 565.2 (calculated 565.2) [M + K]^2+^.

##### Synthesis of PAPTPL-Teg-DA7R

PAPTPL-Teg-N_3_ (23.6 mg, 19.6 µmol, 1 eq.) and DA7R-alkyne (18.3 mg, 19.6 µmol, 1 eq.) were dissolved in tert-butanol:H_2_O:THF 1:1:1 *v*/*v* (450 µL). A mixture of 1.0 M sodium ascorbate in water (19.6 µL, 19.6 µmol, 1 eq.) and 1.0 M CuSO_4_ in water (19.6 µL, 19.6 µmol, 1 eq.) was added. The mixture was stirred for 20 h at room temperature in a sealed vial. The crude product was purified by preparative HPLC (%B: from 60 to 73 in 18 min) to yield PAPTPL-Teg-DA7R (31.9 mg, 14.9 µmol, 76% yield) as a white powder. ESI-MS (*m*/*z*): 1013.5 (calculated 1013.0) [M + H]^2+^, 676.0 (calculated 675.7) [M + 2H]^3+^.

##### Synthesis of PAPTPL-Teg-cA7R

PAPTPL-Teg-N_3_ (21.4 mg, 17.8 µmol, 1.3 eq.) and cA7R-alkyne (12.3 mg, 13.4 µmol, 1 eq.) were dissolved in tert-butanol (300 µL). A mixture of 1.0 M sodium ascorbate in water (12.0 µL, 12.0 µmol, 0.9 eq.) and 1.0 M CuSO_4_ in water (12.0 µL, 12.0 µmol, 0.9 eq.) was added. The mixture was stirred in a microwave reactor (80 °C, 100 W) for 3 min. The crude product was purified by preparative HPLC (%B: from 55 to 80 in 25 min) to yield PAPTPL-Teg-cA7R (13.5 mg, 6.36 µmol, 47% yield) as a white powder. ESI-MS (*m*/*z*): 1004.9 (calculated 1004.5) [M + H]^2+^, 670.3 (calculated 670.0) [M + 2H]^3+^.

### 2.2. Cell Cultures

Pancreatic cancer cell lines (HPAFII, Aspc1, HS766T, PANC1, and BxPc3) were from ATCC (Manassas, VA, USA); Pan02 cells were from the National Cancer Institute (Frederick Cancer Research and Development Center, Frederick, MD, USA). Cells were maintained in Dulbecco’s Modified Eagle Medium (DMEM, Aurogene, Roma, Italy) (Pan02, HPAFII, Aspc1, HS766T, PANC1) or in RPMI medium (BxPC3) supplemented with 10% (*v*/*v*) fetal bovine serum (FBS), 100 U/mL penicillin G, 0.1 mg/mL streptomycin, and 5% glutamine. Cells were grown at 37 °C in a humified atmosphere supplemented with 5% CO_2_.

### 2.3. Protein Extraction

Cells were washed with cold PBS, detached with trypsin, and centrifuged at 2000× *g* for 5 min. The resulting pellet was then resuspended in 0.2 mL RIPA lysis buffer containing protease and phosphatase inhibitors (Merck Life Science S.r.l., Milano, Italy). Each sample was incubated for 30 min on ice and then physically disaggregated with three freeze-and-thaw cycles using liquid nitrogen. The lysates were then centrifuged at 20,000× *g* for 15 min at 4 °C, and finally transferred to a new tube. Quantification of the total protein content in each lysate was performed using the BCA assay kit (Thermo Fisher Scientific, Waltham, MA, USA).

### 2.4. Western Blot

Proteins (30 µg for each sample) were separated by SDS-PAGE (Pre-cast NuPAGE 4–12% Bis-Tris Gels, Life Technologies, Carlsbad, CA, USA). After electrophoretic separation, proteins were transferred to nitrocellulose membranes. Membranes were incubated for 2–3 min at room temperature with a Pounceau Red 0.5% (*w*/*v*) solution in 5% (*v*/*v*) acetic acid (Merck Life Science S.r.l., Milan, Italy) for total protein staining. Images of the Pounceau Red staining were acquired in bright field using a UVITEC Eppendorf apparatus. Membranes were then rinsed with milliQ water and destained with Tris-buffered saline buffer supplemented with 0.1% Tween-20 (TBS-T; Merck Life Science S.r.l., Milan, Italy). The membranes were saturated for 20 min in TBST with 5% nonfat milk (Merck Life Science S.r.l., Milan, Italy) and for 40 min in TBST with 5% bovine serum albumin (Merck Life Science S.r.l., Milan, Italy), and then incubated overnight at 4 °C with the primary antibody. Primary antibodies were anti-neuropilin-1 (Santa Cruz Biotechnology Inc. (Heidelberg, Germany), sc-5307, diluted 1:500 in TBST + 1% milk and 1% BSA) and anti-VEGFR2 (Abcam (Cambridge, UK), ab221679, diluted 1:1000 in TBST + 5% BSA). The following day, secondary anti-mouse (ABclonal (Woburn, MA, USA), AS003) or anti-rabbit (Cell Signaling (Danvers, MA, USA), #7074) antibodies were incubated in TBST 1% nonfat milk and 1% bovine serum albumin for 1 h at room temperature. The chemiluminescence signal was detected using digital imaging by the UVITEC Eppendorf apparatus (Eppendorf, Milan, Italy).

### 2.5. Blood Stability

Procedures involving animals were approved by the University of Padova Ethical Committee for Animal Welfare (OPBA) and by the Italian Ministry of Health (Permit Number 846/2021-PR), and conducted with the supervision of the Central Veterinary Service of the University of Padova, in compliance with Italian Law DL 26/2014, embodying UE Directive 2010/63/EU. Mice were anesthetized and blood was withdrawn from the jugular vein and heparinized. Blood samples (1 mL) were spiked with compound (5 µM; dilution from a 5 mM stock solution in DMSO) and incubated at 37 °C for 4 h (the maximum period allowed by blood stability). Aliquots (100 µL) were taken after 10 min, 30 min, 1 h, 2 h, and 4 h, kept on ice, and treated as follows: blood was mixed with 5 volumes of 0.1%TFA in CH_3_CN, mixed and sonicated for 2 min, and finally centrifuged for 7 min at 12,000× *g*; the supernatant was collected and analyzed by UHPLC. Quantitative analyses were carried out by UHPLC (1290 Infinity LC System, Agilent Technologies, Milan, Italy) using a reversed-phase column (Zorbax RRHT Extend C18, 1.8 µm, 50 × 3.0 mm, Agilent Technologies) and a UV diode array detector (190–500 nm), at a flow rate of 0.6 mL/min. Solvents A and B were 0.1% TFA in water and CH_3_CN, respectively. The gradient for B was as follows: 10% for 1 min, then from 10 to 100% in 8.0 min, 100% for 0.5 min, then from 100% to 10% in 1.5 min. The injection volume was 5 µL. The eluate was preferentially monitored at 312 nm (corresponding to an absorbance maximum of psoralenic compounds). The temperature of the column was kept at 25 °C. Concentration of PAPTPL or its derivatives was determined using the following calibration curve, which correlates peak area at 312 nm (y) with compound concentration in the sample (x): y = 6.3321x. The stability in blood was expressed as % of PAPTPL released relative to the amount of the derivative initially spiked in the sample.

### 2.6. Hemolysis Assay

Hemolysis studies were conducted as described in [48]. Briefly, freshly withdrawn blood samples were centrifuged at 2000× *g* for 10 min to obtain erythrocytes; the pellet was then washed three times with sterile PBS (pH 7.4). Erythrocytes (5 × 10^7^ cells in 1 mL of PBS for each experimental condition) were incubated with the tested compounds at the concentrations of 1, 0.5, 0.25, or 0.125 nmoles/10^7^ cells for 15 min. The samples were then centrifuged and the absorbance (Abs) of the supernatant was measured at 540 nm. Triton X-100 1% and PBS were used as positive and negative control, respectively. The percentage of hemolysis was calculated with the following equation:% hemolysis = [(Abs_sample_ − Abs_PBS_)/(Abs_Triton X-100_ − Abs_PBS_)] × 100

### 2.7. Flow Cytometry

HPAFII and Pan02 cells were seeded in 6-well plates at a density of 300,000 and 150,000 cells/well, respectively, and grown for 48 h. They were then washed twice with PBS, and incubated with a solution of 5 µM FITC-DA7R in DMEM without Phenol Red and serum for 15 min or 2 h. At the end of the incubation, cells were washed, detached with trypsin, pelleted, and resuspended in 1 mL HBSS. Cell suspensions were analyzed with a flow cytometer (BD Biosciences, Franklin Lakes, NJ, USA).

### 2.8. Cell Uptake of PAPTPL-I-DA7R

HPAFII and Pan02 cells were seeded in 6-well plates at a density of 300,000 and 150,000 cells/well, respectively, and grown for 48 h. They were then washed twice with PBS and incubated for 15 min at 37 °C with a solution of 5 µM PAPTPL-I-DA7R in DMEM without Phenol Red and serum. At the end of the incubation, cells were washed twice, 100 µL of PBS were added, and finally the cells were scraped and collected. A volume of 100 µL of CH_3_CN + 0.1%TFA was added, and the mixture was vortexed and sonicated for 5 min. Samples were finally centrifuged for 5 min at 12,000× *g*, and the supernatant was collected and the amount of PAPTPL-I-DA7R was quantified by HPLC/UV, as described in Section 2.5 (blood stability). The remaining cell pellets were resuspended in 200 µL of SDS 2%, and the total protein amount of each sample was quantified with the BCA assay. Cell uptake of PAPTPL-I-DA7R was expressed as nmoles of PAPTPL-I-DA7R/total protein.

### 2.9. Statistics

Significance in comparisons was assessed using the Mann–Whitney–Wilcoxon test.

## 3. Results

### 3.1. Chemistry

We first synthesized a fluorescent derivative of the retro-inverso peptide variant (FITC-DA7R), to follow peptide uptake into PDAC cells. We then proceeded to link the same peptide to the chemotherapeutic compound PAPTP, using a carbamate bond to allow the release of the active drug over time. The first derivative (PAPTPL-I-DA7R) was produced attaching the peptide to the molecule through an Isoleucine linker. An analog comprising a tetraethylene glycol chain, inserted to increase the water solubility of the construct, was then synthesized using click chemistry (PAPTPL-Teg-DA7R). Following the same procedures, we also synthesized an analog of this latter compound comprising the cyclic instead of the retro-inverso peptide (PAPTPL-Teg-cA7R).

#### 3.1.1. Peptide Design and Synthesis

Linear peptides (A7R and DA7R (Figure 2)) were synthesized by manual solid phase peptide synthesis starting from a preloaded 2-Cl-Trytil resin using the Fmoc/HBTU chemistry. To prevent the formation of byproducts due to deletion reactions either in the Pro-Pro or in the Xaa-Pro sequences, double coupling of either Fmoc-Pro-OH or Fmoc-Xaa-OH residue was performed using HATU as a coupling agent.

In the synthesis of cA7R (Figure 2), an additional propargylglycine residue (Pra) was added to the N-terminal amino group of the resin-attached peptide, before its final cyclization. The introduction of this residue provides an alkyne moiety in the side chain of the cyclic peptide to be used for the subsequent condensation reaction to the PAPTP azido analogue (see below). The Pra-A7R peptide was detached from the resin by treatment with 1% trifluoroacetic acid (TFA) in DCM, and the crude peptide was cyclized in a diluted DMF solution (final peptide concentration 1 mM) by the addition of PyBop as a coupling reagent in the presence of HOBt and DIEA. After treatment with TFA to remove side chain protecting groups and RP-HPLC purification, the peptide was obtained in good yield with a purity of >95%.

#### 3.1.2. Synthesis of PAPTPL-I-DA7R

PAPTPL-I (Figure 1) was synthesized according to the procedure described in [47], and then coupled to DA7R with the peptide still attached to the 2-Cl-Trytil resin following standard solid-phase synthetic procedures. After detachment from the resin by TFA treatment, the PAPTPL-I-DA7R conjugate was purified by preparative HPLC, and characterized by ESI-MS (Appendix A).

#### 3.1.3. Synthesis of PAPTPL-Teg-DA7R and PAPTPL-Teg-cA7R

##### Synthesis of PAPTPL-Teg-N_3_

The synthesis of PAPTPL-Teg-N_3_ is illustrated in Figure 1. In the first step, 14-azido-3,6,9,12-tetraoxatetradecan-1-amine was converted to the 4-nitrophenyl carbamate derivative **1** by reaction with bis(4-nitrophenyl) carbonate in the presence of DMAP. In the second step, PAPTPL (synthesized according to the procedure described in [47]) was coupled to the intermediate **1** in the presence of DMAP to afford PAPTPL-Teg-N_3_ in excellent yield (93%). The Teg chain was introduced into the structure of PAPTPL-Teg-N_3_ to improve the water solubility of the resulting derivative.

##### Synthesis of PAPTPL-Teg-DA7R

PAPTPL-Teg-DA7R (Figure 1) was obtained by copper(I)-catalyzed azide-alkyne cycloaddition between PAPTPL-Teg-N_3_ and DA7R-alkyne. The product was purified by preparative HPLC and characterized by ESI-MS (Appendix A).

##### Synthesis of PAPTPL-Teg-cA7R

PAPTPL-Teg-cA7R (Figure 1) was obtained by copper(I)-catalyzed azide-alkyne cycloaddition between PAPTPL-Teg-N_3_ and cA7R-alkyne. The reaction was performed in a microwave reactor to speed up the process. The product was purified by preparative HPLC and characterized by ESI-MS (Appendix A).

### 3.2. Neuropilin-1 and VEGFR2 Expression in PDAC Cell Lines

Since A7R is recognized by NRP-1 and VEGFR2, we evaluated whether the two receptors are expressed in vitro in a panel of PDAC cell lines (five human and one murine). The results obtained by Western blot analysis highlighted a great variability in the expression of NRP-1 and VEGFR2 in the different cell lines (Figure 3A). A PDAC tissue lysate was used as positive control and confirmed high expression levels of both receptors in an orthotopically-implanted Pan02-PDAC. HPAFII cells clearly showed expression of both the receptors, and were thus used for the uptake studies. Cultured Pan02 cells, on the other hand, express low levels of both proteins (near/below the detection limit of Western blot), and were thus used as “negative” controls (Figure 3B,C).

### 3.3. Blood Stability and Hemolysis

Since the derivatives were designed as prodrugs, regeneration of the active compound (i.e., PAPTPL) with suitable kinetics once in circulation is essential for future applications. We thus evaluated blood stability and hemolytic activity of the three PAPTPL derivatives synthesized in this study. PAPTPL-I-DA7R turned out to be the most stable, with about 30% of the conjugate being hydrolyzed within 4 h. PAPTPL-Teg-DA7R and PAPTPL-Teg-cA7R underwent a slightly faster hydrolysis, with about 50% hydrolysis in 4 h (Figure 4).

Previous studies have shown that PAPTP-peptide constructs can be hemolytic [29,47,49], and this can heavily impact their future in vivo applications, since the derivatives are expected to reach the bloodstream upon administration. Hemolysis assays revealed a quite different behavior of the three PAPTPL conjugates: PAPTPL-I-DA7R proved to be completely safe and did not induce any hemolysis, while the addition of Teg caused an increase in the ability of the derivative (PAPTPL-Teg-DA7R) to lyse erythrocytes. Finally, the presence of the cyclic peptide caused a dramatic increase in the hemolysis induced by PAPTPL-Teg-cA7R (Figure 5). cA7R itself was completely ineffective towards erythrocytes. Based on these results, we thus considered for further experiments only PAPTPL-I-DA7R.

### 3.4. DA7R-FITC Cell Uptake

To establish whether DA7R is taken up by cells, we performed flow cytometry experiments using the fluorescent construct FITC-DA7R; fluorescence increase was monitored over time (15 min and 2 h) in HPAFII and Pan02 cells. HPAFII cells showed a significant increase of the fluorescence signal (median of the fluorescence of the cell population) compared to the control. This increase was related to NRP-1 and VEGFR2 expression, since it was significantly different from that observed in cultured Pan02 cells, which express non-detectable levels of these proteins (Figure 6).

### 3.5. Cellular Uptake of PAPTPL-I-DA7R

Uptake of PAPTPL-I-DA7R was evaluated in HPAFII and Pan02 cell lines. Cells were incubated with the conjugate for 15 min, and then medium and cells were collected, extracted and analyzed by HPLC/UV. The results showed that uptake of PAPTPL-I-DA7R by HPAFII cells, which express both NRP-1 and VEGFR2, was roughly twice as much as that by cultured Pan02 (in which the levels of the receptors are under the detection limit) (Figure 7).

The results suggest that conjugation with the DA7R peptide confers a certain selectivity in cell uptake, which is related to the expression levels of NRP-1 and VEGFR2 in the cells.

## 4. Discussion

We have reported the successful development and preliminary testing of a peptide-drug conjugate directed against PDAC, one of the most terrible cancers on the clinical scene. The concept underlying the new molecule is that of a “prodrug”, comprising a targeting portion and an apoptosis-inducing chemotherapeutic drug to be released. The former is based on peptide DA7R, expected to target the NRP-1/VEGFR2 receptor complex in PDAC cells, as it does in other systems. As payload, i.e., chemotherapeutic active principle, we used PAPTP, a mitochondriotropic drug under development in our group [44,45,46,50,51,52].

The segment connecting these two key parts of the molecule comprises a linker needed to separate the hydrolysis-prone carbamate bond system from the activating positive charge of the triphenylphosphonium moiety. We followed, in this respect, the same approach used to build a conjugate of PAPTP and peptide Angiopep-2 [47]. As a labile joint conferring the prodrug character to the construct, we considered using a carboxyester group, but preliminary experiments showed that in blood it was hydrolyzed too rapidly to be useful. The more solid carbamate, which we had already used in other cases (e.g., [47]), was confirmed to be a more suitable choice, hydrolyzing with a t_1/2_ of about 4–6 h (Figure 4).

We also synthesized variants comprising, besides PAPTPL and cA7R or DA7R, a Teg chain, inserted to increase the water solubility of the construct, as well as a triazole ring produced by the coupling of the two halves of the construct by a “click” reaction as a final step in the synthesis. In all the derivatives, the Arg side chain of the peptide was free and not involved in conjugation; given the literature regarding the interaction of DA7R and cA7R with VEGFR2 and NRP-1 [19,20], this is expected to be sufficient to retain a certain affinity for the receptors. Furthermore, the “cargo” (i.e., PAPTPL-I, PAPTPL-Teg) has a flexible hydrophobic structure which could well accommodate structural changes of the construct for an optimal interaction with the receptors. This strategy was already proven to be successful with a DA7R derivative with myristic acid: even if the conjugation involved the N-terminal Arg amino group of DA7R, the derivative was shown to retain (and also have improved) cell uptake and glioma-homing properties compared to DA7R itself [53].

The PAPTP constructs comprising a Teg chain were not utilized in uptake studies because they proved hemolytic in preliminary studies (Figure 5). Their behavior confirms that elaborations, such as the attachment of “cargo” groups or linker moieties, may confer the ability to cause lysis of erythrocytes [47], even though the peptide itself shows no, or little, such activity. Such controls are therefore clearly needed for each construct intended to be transported by the bloodstream.

The results of the uptake experiments validate the idea that DA7R may be a good candidate as a peptide targeting PDAC. The data obtained with the FITC-DA7R and PAPTPL-I-DA7R (Figure 6 and Figure 7) clearly show that the delivery of “cargo” to the cellular interior is strongly favored by the expression of the receptor molecules on the cell surface. Since PDAC cells and the tumoral stroma express high levels of these receptors in comparison with healthy tissue [30,35], this in vitro selectivity may well be reflected in vivo.

So far, A7R has been mainly used to enhance transport across the BBB and anti-glioma/glioblastoma activity. The constructs we developed may therefore be useful also in that field, especially in view of the current understanding of the role of neuropilin in those cancers [24,54].

In principle, A7R might be used to guide to their target the major drugs currently used—with little success—against PDAC, namely, 5-FU and gemcitabine. Indeed, since these are generic “anti-metabolite” drugs, focusing their delivery may be advantageous, also to reduce undesirable side-effects. They are more hydrophilic than PAPTP, thus they may be less problematic than PAPTP constructs from this point of view. These drugs, also including PAPTP and drug combinations such as FOLFIRINOX [3,55], may be steered using conjugates or DA7R-decorated nanovehicles.

## Data Availability

Raw data are available from the authors upon request.

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
