# Peer review of "DA7R: A 7-Letter Zip Code to Target PDAC"

_pharmaceutics, 2023, doi:10.3390/pharmaceutics15051508_

Round 1

Reviewer 1 Report

The article by Sofia Parrasia et al., “DA7R: 7-letter zip code to target PDAC”, describes a proposed new route for drug delivery of PAPTP (one of the promising compounds targeting mitochondria) to cells of pancreatic ductal adenocarcinoma (PDAC) which is the most common type of pancreatic cancer and is among the most aggressive and still incurable cancers. As a drug delivery vector leading PAPTP drug to cancer cells, the authors propose the use of the A7R peptide with affinity for the nauropilin-1 (NRP-1) receptor, and its retro-inverso derivative DA7R and its cyclic head-to-tail form (cA7R). The work contains extensive research material and elements of novelty. In general, the article can be published in Pharmaceutics, however, I have a few minor or major comments.

 Major comments:

1.       It is known from the literature that, according to the C-End-Rule (CendR), the amino acid arginine at the C-terminus of the A7R peptide should be present with a free carboxyl group. Adding another amino acid and even blocking the free carboxyl group of this arginine residue (e.g. by amidation) eliminates the biological activity of the A7R peptide - its affinity for the binding pocket of the NRP-1 b1 subdomain [Peng, K. et al. “Targeting VEGF–neuropilin interactions: A promising antitumor strategy”, Drug Discov. 2019, 24, 656–664.; Teesalu, T. et al. “C-end rule peptides mediate neuropilin-1-dependent cell, vascular, and tissue penetration”, Proc. Natl. Acad. Sci. USA 2009, 106, 16157–16162]. In both tested conjugates (PAPTPL-Teg-DA7R and PAPTPL-Teg-cA7R, Figure 1) the PAPTP drug is linked (via two linkers) to the DA7R or cA7R peptide precisely through arginine residue. The authors are asked to explain/discuss this issue.

2.       In addition, the authors of MasÅ‚owska et al. article (Biomedicines 11, 564 (2023)) showed by ELISA test that the retro-inverso analogue of the A7R peptide (named DR7A in the paper, having a free amino group instead of a free carboxyl group) has practically no affinity for binding with the NRP-1 receptor, which is consistent with the C-End-Rule mentioned above, while in this manuscript such affinity is observed. The authors are asked to explain/discuss this issue.

3.       The authors are asked to inform why mouse blood was used instead of human serum to test the stability of the synthesized conjugates. In the above mentioned work (Biomedicines 11, 564 (2023)), stability studies of DA7R peptide derivatives conducted with the use of human serum showed very low and unacceptable stability of DA7R peptide-based compounds from the point of view of use them in medicine.

 Minor comments:

1.       There is no explanation of the abbreviations Teg and DIPEA in the text at all. The abbreviations TFA and DCM are explained in the line 372, while  they are used much earlier – in the lines 143 and 168, respectively.  I suggest to include abbreviations list into the manuscript.

2.       There are two different abbreviations for neuropilin-1 receptor in the manuscript: NRP-1 and Nrp-1 (in Fig. 3)

3.       The information in lines 481-492 (in the Discussion section) is the same as in the Introduction section.

4.       Lines 554-559 should be removed.

Author Response

We thank the Reviewers and the Editor for their time and their useful comments. We have done our best to comply with the requests. We provide below a point-by-point reply.

Referee #1

The article by Sofia Parrasia et al., “DA7R: 7-letter zip code to target PDAC”, describes a proposed new route for drug delivery of PAPTP (one of the promising compounds targeting mitochondria) to cells of pancreatic ductal adenocarcinoma (PDAC) which is the most common type of pancreatic cancer and is among the most aggressive and still incurable cancers. As a drug delivery vector leading PAPTP drug to cancer cells, the authors propose the use of the A7R peptide with affinity for the nauropilin-1 (NRP-1) receptor, and its retro-inverso derivative DA7R and its cyclic head-to-tail form (cA7R). The work contains extensive research material and elements of novelty. In general, the article can be published in Pharmaceutics, however, I have a few minor or major comments.

 Major comments:

  1. It is known from the literature that, according to the C-End-Rule (CendR), the amino acid arginine at the C-terminus of the A7R peptide should be present with a free carboxyl group. Adding another amino acid and even blocking the free carboxyl group of this arginine residue (e.g. by amidation) eliminates the biological activity of the A7R peptide - its affinity for the binding pocket of the NRP-1 b1 subdomain [Peng, K. et al. “Targeting VEGF–neuropilin interactions: A promising antitumor strategy”, Drug Discov. 2019, 24, 656–664.; Teesalu, T. et al. “C-end rule peptides mediate neuropilin-1-dependent cell, vascular, and tissue penetration”, Proc. Natl. Acad. Sci. USA 2009, 106, 16157–16162]. In both tested conjugates (PAPTPL-Teg-DA7R and PAPTPL-Teg-cA7R, Figure 1) the PAPTP drug is linked (via two linkers) to the DA7R or cA7R peptide precisely through arginine residue. The authors are asked to explain/discuss this issue.

The C-end rule (CendR) sequence motif  is [R/K]XX[R/K]. This sequence is not present in the A7R peptide (ATWLPPR), which however contains an LPPR sequence, which was demonstrated to be crucial (expecially the C-terminal arginine) for the inhibitory effect of A7R on VEGF165 binding to NRP-1 (Starzec et al, Peptides 2007, 28, 2397-2402).

A7R is known to be recognized not only by NRP-1, but also by VEGFR2 (Binétruy-Tournaire, Demangel et al. 2000; Lu, Chen et al. 2019); in this case, peptide-receptor interactions mainly involve the Trp residue, but also Leu and Pro, while Arg locates at the outer edge of the binding pocket (Lu, Chen et al. 2019).

Protease-resistant A7R variants (DA7R, having a DArg at the N-terminal, and cA7R, in which cyclization involves the COOH of Arginine) were demonstrated to retain affinity for both VEGFR2 and NRP-1, as measured by Surface plasmon resonance assays and  molecular docking analyses (Ying M et al, ACS Mater.Interfaces 2016,  8, 13232-13241; Ying M et al, J Controlled Release 2016, 243, 86-98), despite the absence of a C-terminal Arg or the involvement of Arg COOH in cyclization.  According to these results, it is expected that binding of PAPTPL to DA7R or cA7R as described in this manuscript  would be sufficient to retain a certain affinity for the receptors, since Arg side chain is left free in all cases. Furthermore, the “cargo” (i.e., PAPTPL-I, PAPTPL-Teg) has a flexible and hydrophobic structure, which could well accommodate structural changes of the construct for an optimal interaction with the receptors. This strategy was already proved to be successful with a  DA7R derivative in which myristic acid was conjugated to the N-terminal amino group of Arg in DA7R; the construct was shown to retain (and also have improved) cell uptake and glioma-homing properties compared to DA7R itself (Ying M et al, ACS Mater.Interfaces 2018, 10, 19473-19482).

We added text discussing these aspects in the Introduction (page 2) and Discussion (page 15).

  1. In addition, the authors of Masłowska et al. article (Biomedicines 11, 564 (2023)) showed by ELISA test that the retro-inverso analogue of the A7R peptide (named DR7A in the paper, having a free amino group instead of a free carboxyl group) has practically no affinity for binding with the NRP-1 receptor, which is consistent with the C-End-Rule mentioned above, while in this manuscript such affinity is observed. The authors are asked to explain/discuss this issue.

        The results by MasÅ‚owska et al. are quite surprising and different from data obtained by previous studies; in the papers by Ying M et al (Ying M et al, ACS Mater. Interfaces 2016, 8, 13232-13241), for example, Authors performed Surface plasmon resonance assays and demonstrated that DA7R has high binding affinity to both VEGFR2 and NRP-1 (KD values of 8.41 and 2.31 nM, respectively), similar to those obtained with A7R. This would be more in accordance with the rationale of retro-inverso peptides, in which the amino acids sequence is reversed and the chirality of each residue is inverted; in this way, the retro-inverso peptide and the native peptide have similar chemical topology with respect to their side chains, allowing the recognition of the same biological targets by the two peptides.

  1. The authors are asked to inform why mouse blood was used instead of human serum to test the stability of the synthesized conjugates. In the above mentioned work (Biomedicines 11, 564 (2023)), stability studies of DA7R peptide derivatives conducted with the use of human serum showed very low and unacceptable stability of DA7R peptide-based compounds from the point of view of use them in medicine.

        Conjugates were tested in mouse blood, and not in human serum, because mice represent the next step for testing their performances in vivo and because studies with human samples would require ethical approvals that we don’t have. 

In the above mentioned work (Biomedicines 11, 564 (2023)), stability studies of DA7R peptide derivatives in human serum showed a rapid disappearance of the HPLC peak relative to the derivative, and the appearance and increase during time of another HPLC peak with lower retention time. However, this new peak has not been characterized. If proteolytic cleavage takes place, one would expect to detect multiple HPLC peaks corresponding to the progressive erosion of the peptide chain during time.

According to our data, the hydrolysis of the carbamate bond is expected to be faster (independently from using human or mouse serum) than hydrolysis of the peptide chain.

 Minor comments:

  1. There is no explanation of the abbreviations Teg and DIPEA in the text at all. The abbreviations TFA and DCM are explained in the line 372, while  they are used much earlier – in the lines 143 and 168, respectively.  I suggest to include abbreviations list into the manuscript.

        Thank you for the suggestion. We now included a list of Abbreviations at the end of the manuscript.

  1. There are two different abbreviations for neuropilin-1 receptor in the manuscript: NRP-1 and Nrp-1 (in Fig. 3)

Sorry for the error; we now corrected the abbreviation in Fig.3, so that only the abbreviation NRP-1 is used throughout the text

  1. The information in lines 481-492 (in the Discussion section) is the same as in the Introduction section.

        We repeated the information (with different words) in the Introduction and then at the beginning of Discussion to remark and resume the key aspects of the study. We now shortened the text in the Discussion.

  1. Lines 554-559 should be removed.

According to the pdf version of the manuscript (“pharmaceutics-2361190-peer-review.pdf”), these lines correspond to text about Author contributions and Fundings. We don’t understand the suggestion.

Author Response

We thank the Reviewers and the Editor for their time and their useful comments. We have done our best to comply with the requests. We provide below a point-by-point reply.

Referee #2 

The manuscript „DA7R: a 7-letter zip code to target PDAC “ by Parrasia et al. is well written and organized, and the results are clearly presented supporting the given conclusion. This manuscript adds another milestone in the field of peptide medicinal chemistry, specifically in the field of peptide-drug conjugates with anticancer properties. The presented preliminary results look very promising offering a possibility of new approaches to be taken in the treatment of the currently incurable PDAC. After implementing some minor suggestions and comments that I have (please see below) I recommend acceptance in the present form:

  1. In the section 2.1.4 (Synthesis of PAPTPL-I-DA7R) you are mentioning that the synthesis was performed using HATU/DIPEA activation (lines 186-188). It was surprising to me to see that with this method you have used 1h stirring before adding the mixture to the peptidyl (2-Cltrytil) resin followed by overnight reaction. It is known that this method is normally used for short couplings due to the fact that the formation of active guanidinium esters is very fast and with very short duration (around 30 min). Could you please comment a bit on this? Also, I would like to know the motivation behind choosing this type of activation over, let’s say, carbodiimide or phosphonium?

HATU is a coupling reagent used in a 1:1 ratio with the carboxylic group to be condensed. The latter is usually used in a molar excess of  3 or 4 times in respect to the amino group of the peptide to be linked; under these conditions, reaction usually requires short times, as stated by the Reviewer. In the synthesis of PAPTPL-I-DA7R, however, we set this molar ratio to 1:1 because PAPTPL-I (the moiety with the carboxylic acid to be condensed) is very expensive from a synthetic point of view; for this reason, we prolonged the reaction times. Other coupling reagents can obviously be used according to the procedures of each laboratory: the presence of the HOAt is fundamental for the possibility of double anchimeric assistance compared to other commonly used auxiliary nucleophiles.

  1. Line 253, [M - CF3COO + H]2+ if I understood correctly, it means that you are observing a doubly charged ion without a TFA adduct?

Yes, this is correct. The measured m/z value corresponds to the protonated compound (which is also positively charged because of  the triphenylphosphonium moiety; there are therefore 2 positive charges in total) without the trifluoroacetate counter ion (CF3COO-). We have now deleted the “- CF3COO” annotation for clarity, leaving only “M” (intended as the derivative without any counter ions) and any ions associating to it during electrospray ionization.

  1. Line 250, you are mentioning that the click reaction was performed at room temperature for 20h, while the same reaction when cA7R is used is performed in a microwave reactor. I am curious to know if there was any particular reason to use microwave activation in this case?

Each click reaction was conducted with different procedures (room temperature, microwave activation). For each derivative, the conditions giving the best yields are reported in the manuscript.

  1. Line 307, please put a colon after “follows”, line 325, please correct “thrice” with twice.

Thank you. We now corrected lines 307and 325.

  1. In the section 3.5, you are comparing the uptake of PAPTPL-I-DA7R in two different cell lines. Did you also compare the uptake of PAPTPL-I-DA7R vs PAPTPL-I as additional proof of your concept and conclusion that the uptake is due to selective recognition of NRP-1 and VEGFR2 receptors by A7R?

We did not compare uptake of PAPTPL-I-DA7R with that of PAPTPL because these two compounds are expected to have different uptake mechanisms (the former is expected to exploit receptor-mediated transcytosis, the latter is taken up by passive diffusion and then accumulated into mitochondria thanks to the triphenylphosphonium moiety). PAPTPL cell uptake depends on both the plasma membrane and mitochondria transmembrane potential, which could be different among the cell lines we used. Thus, a comparison between PAPTPL-I-DA7R and PAPTPL uptake could give misleading information. Also testing cell uptake of PAPTPL-I could be misleading, because the presence of Isoleucine could lead to an uptake mediated by aminoacid transporters (Es: Azzolini M. et al, Eur J Pharm Biopharm 2017, 115, 149-158), whose expression is not known in the cells we used. Similar uptake differences between cells expressing greatly different levels of VEGFR2 and NRP-1 were obtained with two very different  DA7R derivatives (with very different cargos, i.e., FITC or PAPTPL-I), through different experimental techniques (i.e., flow cytometry and fluorescence detection for FITC-DA7R; HPLC analysis with UV detection for PAPTPL-I-DA7R); this could represent a sufficient evidence that the derivatives are taken up through NRP-1 and VEGFR2 receptors thanks to the presence of DA7R.

  1. Line 513, please give a reference after “elaborations”.

Thank you for the suggestion. We now inserted a reference (Parrasia, Rossa et al, Pharmaceuticals 2021) about a similar hemolytic behaviour by a TAT-PAPTPL derivative.